# Effectiveness of a Novel Food Composed of Leucine, Omega-3 Fatty Acids and Probiotic *Lactobacillus paracasei* PS23 for the Treatment of Sarcopenia in Elderly Subjects: A 2-Month Randomized Double-Blind Placebo-Controlled Trial

**DOI:** 10.3390/nu14214566

**Published:** 2022-10-30

**Authors:** Mariangela Rondanelli, Clara Gasparri, Gaetan Claude Barrile, Santina Battaglia, Alessandro Cavioni, Riccardo Giusti, Francesca Mansueto, Alessia Moroni, Fabrizio Nannipieri, Zaira Patelli, Claudia Razza, Alice Tartara, Simone Perna

**Affiliations:** 1IRCCS Mondino Foundation, 27100 Pavia, Italy; mariangela.rondanelli@unipv.it; 2Unit of Human and Clinical Nutrition, Department of Public Health, Experimental and Forensic Medicine, University of Pavia, 27100 Pavia, Italy; 3Endocrinology and Nutrition Unit, Azienda di Servizi alla Persona ‘‘Istituto Santa Margherita’’, University of Pavia, 27100 Pavia, Italy; gaetanclaude.barrile01@universitadipavia.it (G.C.B.); alessandro.cavioni01@universitadipavia.it (A.C.); francesca.mansueto01@universitadipavia.it (F.M.); alessia.moroni02@universitadipavia.it (A.M.); zaira.patelli01@universitadipavia.it (Z.P.); claudia.razza01@universitadipavia.it (C.R.); alice.tartara01@universitadipavia.it (A.T.); 4Medical Research Abiogen Pharma, 56121 Pisa, Italy; santina.battaglia@abiogen.it (S.B.); riccardo.giusti@abiogen.it (R.G.); fabrizio.nannipieri@abiogen.it (F.N.); 5Department of Biology, College of Science, University of Bahrain, Sakhir Campus, Zallaq P.O. Box 32038, Bahrain; simoneperna@hotmail.it

**Keywords:** sarcopenia, leucine, omega-3 fatty acids, probiotic, *Lactobacillus paracasei* PS23

## Abstract

Sarcopenia is a complex process characterized by a progressive decrease in muscle mass and strength. Various nutrients have been shown to be effective in supporting muscular performance. This randomized clinical trial aimed to evaluate the effectiveness of a 2-month administration of food for special medical purposes composed of omega-3 fatty acids (500 mg), leucine (2.5 g), and probiotic *Lactobacillus paracasei* PS23 (LPPS23), on appendicular lean mass (ALM), muscle performance, inflammatory status, and amino acid profile in sarcopenic patients. A total of 60 participants (aged 79.7 ± 4.8 years and a body mass index of 22.2 ± 2.1 kg/m^2^) were enrolled and randomly assigned to either intervention (*n* = 22) or placebo group (*n* = 28). Comparing the differences in effects between groups (intervention minus placebo effects), ALM increased significantly in the intervention group (*p* < 0.05), with no discernible change in the placebo group. Similarly, significant differences were also observed for the Tinetti scale (+2.39 points, *p* < 0.05), the SPPB total score (+2.22 points, *p* < 0.05), and the handgrip strength (4.09 kg, *p* < 0.05). Visceral adipose tissue significantly decreased in the intervention group compared to the placebo group at 60 days −0.69 g (95% CI: −1.09, 0.29) vs. 0.27 g (95% CI: −0.11, 0.65), groups difference −0.96 (95% CI: −1.52, 0.39, *p* = 0.001). A statistically significant increase in levels of valine, leucine, isoleucine, and total amino acid profiles was observed in the intervention group compared with the placebo group at 60 days (*p* = 0.001). When taken together, these beneficial effects may be attributed to the innovative composition of this special medical-purpose food which could be considered for the treatment of sarcopenia in the elderly.

## 1. Introduction

The population worldwide is continuously growing due to rapid socioeconomic and health transitions, which have led to an increased mean life span [1]. This has led to an increased frequency of age-related diseases, such as cardiovascular disease, diabetes, cancer, osteoporosis, and the less commonly known sarcopenia [2,3]. Sarcopenia is a complex condition characterized by decreased muscle mass, strength, and structural alteration, with increased fat mass (FM) [4,5]. The prevalence of this age-related syndrome is estimated to affect between 5% to 20% of individuals aged 60–70 years, as many as half of those aged 80 or above [6,7,8,9].

Sarcopenic status is deeply connected to the pathological alterations typically associated with old age, such as reduction in postural stability, bone status impairment, changes in glucose homeostasis, and reduction in basal metabolic rate [10,11]. 

Reduced muscle mass and strength have been linked to loss of functional capacity and disability, increased mortality and other adverse outcomes [12]. Consequently, sarcopenia is heavily correlated to economic and social conditions, as evident by the US healthcare expenditure for sarcopenia and sarcopenia-related diseases, comprising 1.5% of the total health expenditure for the year 2000 [13]. Despite these significant economic and social issues, the early detection and intervention of this disease may be crucial in improving outcomes in these patients. 

Diet and nutritional supplementation may represent a valid strategy to help maintain muscle mass and function and combat sarcopenia in the elderly. Nutritional treatment currently recommended is based on adequate dietary protein and amino acid intake with vitamin D supplementation [7,14,15]. 

Aging is associated with sedentary behavior, inflammation, and oxidative stress; these factors are thought to cause anabolic resistance [16,17]. However, supplementation with branched amino acids, including leucine (which are well known for their anabolic effect), has shown promising results in treating sarcopenia and sarcopenia-like models [18,19,20]. Furthermore, leucine has been shown to modify protein turnover in skeletal muscle, decrease proteolysis and increase protein synthesis [20,21], and enhance muscle glucose uptake and metabolism [22]. Other nutrients, such as omega-3 fatty acids, have also been shown as effective in supporting muscular performance with its supplementation being a potentially protective factor against muscle loss and an activator of muscle synthesis [23]. Moreover, the positive effect of specific omega-3 fatty acids such as eicosapentaenoic acid (EPA) and docosahexaenoic acid (DHA) on skeletal muscle has been demonstrated in several animals and in vitro models [24,25,26] as well as in human studies [12,27]. In fact, there is evidence demonstrating that all omega-3 polyunsaturated fatty acids have an anabolic effect on muscle [28] that is independent of their well-known anti-inflammatory effects [29].

Probiotic supplementation is also used to modulate gut microbiota. According to the “Gut-muscle Hypothesis” [30], the intestinal microbiota could act as a mediator between nutrition and the aging phenotype because it has an active role in the regulation of immune cell function, metabolic balance, insulin sensitivity, and gene expression of the host [31,32]. Recent studies have shown a correlation between gut microbiota composition and variability and physical performance in the older population [33,34]. In particular, the probiotic *Lactobacillus paracasei* PS23 (LPPS23) has been studied in animal models in which aged rats supplemented with the probiotic showed decelerated age-related muscle loss [35] and age-related cognitive decline [36].

The primary aim of this randomized clinical trial was to evaluate the effectiveness of a special medical food composed of omega-3 fatty acids, leucine, and the probiotic LPPS23, compared to placebo, on appendicular lean mass (ALM) in sarcopenic patients using dual-energy X-ray absorptiometry (DXA). A secondary aim was to evaluate the effectiveness of this treatment on: (a) physical performance using the short physical performance battery (SPPB), (b) walking and risk of fall using the Tinetti scale, (c) force of hand flexor muscles using the handgrip, (d) changes in body composition using DXA, (e) functional state using the activity daily living (ADL) scale and Barthel index, (f) quality of life using the 12-item Short-form Health Survey (SF-12), (g) mood using the geriatric depression scale, (h) level of inflammation by measuring the C-reactive protein (CRP), fasting blood glucose (FBG), liver and kidney function, and (i) the plasma amino acid profile.

## 2. Materials and Methods

### 2.1. Standard Protocol Approval, Registration, and Patient Consent

This study was approved by the Ethics Committee of the University of Pavia, Italy (approval number 202000070742) and complied with the ethical standards as laid down in the 1964 Declaration of Helsinki with written informed consent obtained from all patients entering the pre-treatment phase. This study was registered under ClinicalTrials.gov (accessed on 19 September 2022) (NCT04702087). 

### 2.2. Study Design and Sample Size

This was a randomized (1:1), double-blind, placebo-controlled, parallel-group, 2-month clinical intervention study undertaken from January to September 2021. The study was conducted at the Geriatric Physical Medicine and Rehabilitation Division of Santa Margherita Hospital, Azienda di Servizi alla Persona, Pavia, Italy. Allocation to the intervention groups was performed via a computer-generated random blocks randomization list, and random assignments were concealed in sealed envelopes.

The sample size calculation was based on the reference study of Logan et al. [37], and the sample size on the primary outcome, as an increase in lean mass of 0.6 kg in the placebo group and +1.6 kg in the intervention group with omega-3 fatty acid (+1 kg increase in lean mass in the intervention compared to placebo) as a percentage (+3.97% omega-3 group and +1.51% placebo) “between groups” (+2.5% increase in lean mass) on treatment variable for 3 months of treatment (continuous variable). Considering two balanced groups with 1:1 ratio allocation (n1 = n2), an effect size of 0.5, an alpha significance level set at 0.05, a dropout rate of 10%, and 80% power in detecting differences between groups, it was estimated that a sample size of 54 patients (27 patients per arm) would be needed for enrollment.

### 2.3. Participants

The study subjects were sarcopenic patients with a diagnosis made according to the revised European Consensus on Definition and Diagnosis criteria [7]. We enrolled male and female subjects aged ≥55 and with a body mass index (BMI) between 20 and 30 kg/m^2^. Subjects with the following conditions were excluded from the study: severe kidney disease (glomerular filtration rate <30 mL/min), moderate-to-severe hepatic failure (Child-Pugh Class of B or C), endocrine diseases associated with disorders of calcium metabolism (with the exception of osteoporosis), psychiatric disorders, cancer (in the previous 5 years), or hypersensitivity to any investigational food component for special medical purposes and subjects taking protein/amino acid supplements (up to 3 months prior to the study). Patients not capable of taking oral therapy and those receiving or having an indication for artificial nutrition or included in another clinical nutrition trial were also excluded from the study. The investigator’s professional judgment on the willingness or capacity of the subject to adhere to protocol requirements was also considered as an exclusion criterion. 

### 2.4. Nutritional Assessment and Nutritional Interventions

Each participant was given an individualized dietary program. A personalized nutritional schedule was prepared for patients in both groups that provided 1.5 g of protein per kg of body weight per day. Calorie intake was evaluated by a trained dietitian with the estimated basal metabolic rate using the Harris–Benedict formula multiplied by the estimated activity factor and protein intake. The dietary scheme consisted of about 55% of carbohydrates and 30% of lipids. In addition, weight history was evaluated by observing any weight loss compared to the usual weight and during the 6 months prior to the baseline visit.

The compliance with the dietary intervention was assessed through the 24 h-dietary recall once a month for three months.

Moreover, during follow-up calls conducted by study dietitians and the in-person dietitian consultation once a month, all participants were asked: “How well have you been following your diet plan? On an analogic scale of 0 to 10, with zero being not at all, four being somewhat, and ten being following the plan very well, where would you place yourself?”. Similarly, the same questions were asked for adherence to physical activity prescriptions. Self-rated adherence scores assisted with identifying participants’ barriers to change and setting personal diet and physical activity goals to achieve by the next follow-up call. Responses to self-rated adherence scores across the 12-week study were averaged for each person, and participants were divided into a high or low level of adherence, split by median score (7.5 for diet and 7.6 for physical activity). The subjects with a low level of adherence have been excluded. 

Subjects were randomly allocated to receive once daily the experimental formula: omega-3 fatty acid (500 mg, consisting n 64.71% EPA, 29.41% DHA and the remaining 5.88% omega-3 in general), leucine (2.5 g), probiotic LPPS23 (“30 Billion”, freeze dried) (OLEP), or the control formula: isocaloric placebo with the same flavor (Figure 1). The experimental and the control formulas were delivered as indistinguishable water-dispersible powder.

The compliance of subjects with intake of nutritional interventions was monitored by inputting the number of daily servings consumed in a diary.

### 2.5. Anthropometric Measurements 

Anthropometric measurements were assessed at the beginning of the study (at baseline; (t0) and after 60 days (t2). Body weight and height were measured following a standardized technique [38], and BMI was then calculated. Abdominal circumference was also evaluated. All anthropometric parameters were measured by the same investigator. 

### 2.6. Body Composition Assessment

Body composition represented by fat-free mass (FFM) and FM were measured using a Lunar Prodigy DXA (GE Medical Systems, WI, USA). The in vivo coefficients of variation were 0.89% for the whole FM and 0.48% for FFM. Visceral adipose tissue (VAT) volume was estimated using a constant correction factor of 0.94 g/cm^3^. This software automatically places a quadrilateral box that represents the android region outlined by the iliac crest, with a superior height of 20% of the distance from the top of the iliac crest to the base of the skull [39]. The calculation of ALM was based on the sum of the fat-free mass from arms and legs. ALM was standardized by BMI according to previous studies [40,41] in order to normalize the data by height and weight. Measurements were performed at baseline (t0), after 30 days (t1) and after 60 days (t2).

### 2.7. Muscle Strength Evaluation

Muscle strength was measured by handgrip strength according to standard procedures by a hydraulic hand dynamometer (Jamar 5030 J1, Sammons Preston Rolyan, Bolingbrook, Illinois, USA, with an accuracy of 0.6 N). The subject holds the dynamometer in the hand to be tested, with the arm at right angles and the elbow by the side of the body, applying an isometric contraction. Evaluation of muscle strength was performed at t0, t1, and t2.

### 2.8. Functional Status Evaluation

Two different tests were used for a correct evaluation of functional status. These included the Barthel Index, which covers all aspects of self-care independence in normal daily living activities, including transfer, walking, stairs, toilet use, dressing, feeding, bladder, bowel, grooming, and bathing, and a score ranging from 0 (completely dependent) to 100 (totally independent) [42] and the ADL score [43]. Both of these evaluations were performed at t0 and t2.

### 2.9. Physical Performance Assessment

Physical performance was assessed using the a) SPPB, which comprised of gait speed, chair–stand test, and the timed up and go test (that assesses the time taken to rise from an armchair, walk 3 m, turn, walk back, and sit down again [44]), and balance (three different tests that assess the ability to stand with the feet together in the side-by-side, semi-tandem, and tandem positions); each component was scored from 0 (not possible) to 4 (best performance), and the scores were added up to a total score ranging from 0 to 12 [45] and b) the Tinetti scale, that measures characteristics associated with falls, assessing balance (14 items; 24 points), and gait (10 items; 16 points) for a total score up of 40 (the higher the score, the better the performance) [46]. This assessment was performed at t0 and t2.

All patients performed specific personalized endurance and aerobic physical activity training. An individualized, moderate-level (Borg Rate of Perceived Exertion scale score of 12–14) physical fitness and muscle mass-promoting program was set up for all in-patients [47]. All exercise sessions were supervised by trained staff, where the monitoring of the individual exercise ability of each patient and eventual adjusting of the intensity level was performed, as appropriate. The physical assessment consisted of daily exercise sessions performed five times a week. Initial sessions had a duration of 20 min, gradually increasing with the intensity of the exercises, up to 30 min. All sessions comprised the following: a warm-up period of 5 min, a progressive sequence of 5 to 10 min from seated to standing muscle-strengthening exercises: toe raises, heel raises, knee lifts, knee extensions (in the seated position); hip flexions and lateral leg raises (standing next to a chair for stability); ankle-weight bearing exercises, with weights ranging from 0.5 to 1.5 kg according to each participant’s strength as resistance increased progressively; hip and leg extensions (using resistance bands). Upper-body exercises were also carried out and consisted of double-arm pull downs and bicep curls. Patients were then asked to repeat exercises up to eight times, as appropriate; 5 to 10 min balance and gait exercises: one-leg stands, tandem stands, multidirectional weight shifts, tandem walk, as well as practicing proper gait mechanics focusing on balance maintenance and increasing stride length, while changing direction and/or gait pattern, and ending the session with a cool-down period of 5 min. The minimum duration of the physical intervention program was 4 weeks but could be extended up to 8 weeks if needed. The decision to complete the rehabilitation and to discharge the patient was taken by a multidisciplinary team (geriatrician, physiatrist, physiotherapist, and nurse) once the duration of each exercise session was stabilized to 30 min, and no increase in exercise intensity could be undertaken for five consecutive days.

### 2.10. Quality of Life Assessment

The quality of life of participants was assessed using the 12-item Short Form Survey (SF-12) health survey. This consisted of a short, generic health-status measure reproducing the 2 summary scores of the SF-36 (physical component summary score and mental component summary score) that measures eight health domains: physical functioning, role-physical, bodily pain, general health, vitality, social functioning, role-emotional, and mental health [48]. This questionnaire was completed at t0 and t2.

### 2.11. Mood Assessment

The depression status of patients was evaluated using the 30-item Geriatric Depression Scale. This scale represented a reliable and valid self-rating depression screening scale for elderly individuals [49] and was performed at t0 and t2.

### 2.12. Evaluation of Blood Pressure

Blood pressure was measured with patients in the sitting position after a 5 min rest at t0 and t2.

### 2.13. Biochemical Parameters

All biochemical parameters were examined before the start of the study at baseline t0 and t2. Venous blood samples were drawn after an overnight fast. In order to avoid venipuncture stress, blood samples were obtained through an indwelling catheter inserted in an antecubital vein. The concentrations of free essential amino acids leucine, isoleucine, and valine in plasma samples were measured using the AminoQuant II amino acid analyzer based on the HP 1090 HPLC system (SpectraLab Scientific Inc., Markham, ON, Canada) with fully automated pre-column derivatization using ortho-phthalaldehyde (OPA) and 9-fluorenylmethyl-chloroformate (FMOC) reaction chemistry as specified by the manufacturer. Amino acids were detected by measuring UV absorbance at 338 and 262 nm and analyzed as follows: 2 mL plasma samples were de-proteinized by adding 500 µL of 0.5 N HCl, and after centrifuging at 5000× *g* for 10 min at 5 °C the supernatant was concentrated to 200 µL under a nitrogen stream then further filtered on a 0.45 µm Millipore filter. Aliquots (1 µL each) were automatically transferred to the reaction coil and derived with the reagents listed above. The remaining de-proteinized serum was stored at −20 °C. Analysis of samples was performed in duplicate, and values reported for each amino acid are the mean of two independent measurements. The mean level of the lowest detectable measurement of amino acid was 3–5 pmol/µL of material injected. Levels of amino acid concentrations were expressed as moles/l. FBG level was measured using the automatic biochemical analyzer (Hitachi 747, Tokyo, Japan). CRP was determined by Nephelometric High-Sensitivity CRP (Dade Behring, Marburg, Germany).

### 2.14. Safety Assessment and Monitoring of Adverse Events

The following routine blood biochemistry parameters of liver and kidney function were evaluated using enzymatic-colorimetric methods: alanine aminotransferase, aspartate aminotransferase, gamma glutamyl transferase, and creatinine. These parameters were evaluated at the start and at the end of treatment.

The presence of adverse events was reported by subjects as well as open-ended inquiries by members of the research staff. 

### 2.15. Statistical Analysis

Statistical analysis and the reporting of this study were conducted in accordance with the Consolidated Standards of Reporting Trials (CONSORT) guidelines, with the primary analysis based on the full analysis set. For baseline variables, data are presented as frequencies and proportions for categorical data and mean and standard deviation (SD) for continuous variables. A comparison of baseline variables was performed using a Chi-square test for categorical outcomes and unpaired t-tests for continuous variables. 

In the primary analysis, the baseline-adjusted means and 95% confidence interval (CI) estimated by analysis of covariance (ANCOVA) with the change in primary outcomes were compared intra-group and between the placebo and intervention groups (intervention–placebo). Comparisons were adjusted for age, gender, and baseline values. ANCOVA were used for the secondary outcomes at each time point. All *p*-values were two-sided, and a *p*-value of ≤0.05 was considered statistically significant. Spearman’s correlations were assessed on pre- post mean changes in the intervention and placebo groups. A percentage standardization of the visual analogic scale for diet and physical activity compliance was performed. Qualitative variables were described as frequencies (%) with respect to an ideal situation (100% compliance), and the statistical differences were evaluated by Chi-squared tests.

All analysis was performed using SPSS statistical software (version 21.0, SPSS Inc., Chicago, IL, USA).

## 3. Results

A total of 60 patients were enrolled and randomly assigned to either the placebo or intervention group, as shown in the flowchart of the study (Figure 2). The primary and secondary outcomes were analyzed on the full analysis set. Baseline clinical characteristics were similar between the two groups (Table 1). The mean age was 79.71 ± 4.84 years, and mean BMI was 22.27 ± 2.12 kg/m^2^. In according to Tinetti, all patients had a moderate-high risk of fall, as reflected by a score of 17.33 ± 3.84.

Table 2 shows the data of the primary and secondary outcome variables at 60 days post-baseline. An increase in ALM was observed in the intervention group that failed to achieve statistical significance, but in the placebo group, a statistically significant decrease in ALM was observed. The change in all functional tests in the intervention group was significantly higher overall than in the placebo group. Additionally, when comparing the effects between groups (intervention minus placebo effects), Significant differences in the Tinetti scale (+2.386 points; 95% CI: 1.054, 3.719), the SPPB total score (+2.219 points; 95% CI: 1.436, 3.002), and the handgrip strength test (4.087 kg; 95% CI: 2.781, 5.392) were observed. The results also showed significant increases (*p* < 0.001) in body weight, BMI, and waist circumference in the intervention group compared to the placebo group after 60 days of treatment. 

The plasma inflammation marker CRP significantly decreased in the intervention group compared to the placebo group at 60 days −0.690 (95% CI: −1.094, −0.286) vs. 0.266 (95% CI: −0.113, 0.645), groups difference −0.956 (95% CI: −1.522, 0.390 *p* = 0.001). There were no significant differences between the two groups for all other biochemical markers examined and for compliance with diet and physical activity.

Figure 3 shows the changes occurred in body composition changes from the start of the treatment and after 30 and 60 days of treatment.

Weight, FM, and ALM were measured after 30 days from the start of the treatment; no significant differences between OLEP and placebo group were reported in the observed variables. These variables showed statistically significant trend changes at the end of the treatment period.

Figure 4 depicts the amino acids profile mean difference changes from baseline to 60 days. Amino acids profile represented by valine, leucine, isoleucine, and total amino acids showed a statistically significant increase in the intervention group in comparison to the placebo group at 60 days (*p* = 0.001).

Regarding safety, one adverse event was recorded during the study but not related to the intervention. The food given for special medical purposes administered was well tolerated.

## 4. Discussion

The present study demonstrated that a 2-month treatment with food for special medical purposes based on omega-3 fatty acids, leucine, and the probiotic LPPS23 is effective in the improvement of ALM, in addition to all of the muscular functional tests carried out. Specifically, comparing the between-groups effects (intervention minus placebo effects) demonstrated significant increases in the Tinetti scale (+2.386 points), the SPPB total score (+2.219 points), and handgrip strength (4.087 kg). The results also showed significant increases in body weight, BMI, and waist circumference in the intervention group compared to the placebo group over the 60 days treatment period. Moreover, the intervention prevented the loss of ALM that was, instead, observed in the placebo group. 

In our study, no significant difference was observed between the two groups with regards to body weight, FM, and ALM after 30 days of intervention, whereas 60 days of treatment were required to allow us to observe statistically significant differences. These improvements could be attributed to the innovative composition of the food for special medical purposes administered. In fact, the experimental formula used in this study contains 500 mg of omega-3 fatty acids, 2.5 g of leucine, and the probiotic LPPS23. After controlling for BMI in our analysis, ALM did not change statistically within and between the two groups.

It is recognized that leucine is not only a component amino acid of proteins but also has anabolic and anticatabolic functions. Several studies have demonstrated the ability of leucine to modify protein turnover in skeletal muscles, thereby decreasing the proteolysis rate and increasing protein synthesis [20,21]. It is currently recommended that an intake of 3 g of leucine at the three main meals, together with 25–30 g of protein, is necessary in order to prevent or recover the loss of lean mass in the elderly [50]. Moreover, Hun et al. [51] reported that exercise together with a leucine-rich essential amino acid mixture supplementation at a dose of 3 g twice a day for 3 months might be effective in enhancing not only muscle strength but also combined variables of muscle mass and walking speed and of muscle mass and strength in sarcopenic women [51]. Furthermore, a recent randomized, placebo-controlled trial demonstrated that the daily administration of 6 g of leucine significantly improved some features of sarcopenia, such as functional performance measured by walking time and improved lean mass index in elderly individuals [52]. 

The present study measured plasma levels of total amino acids and that of leucine, isoleucine, and valine at baseline and after 2 months of treatment. The findings revealed a statistically significant increase in the intervention group compared with the placebo group. Corroborating our observations, in a pilot study by Tosukhowong and colleagues, plasma levels of branched-chain amino acids and essential amino acids were significantly higher in whey-supplemented Parkinson’s disease patients after 6 months compared to the baseline values [53]. We have recently published a systematic review showing that daily supplementation with the omega-3 fatty acids EPA and DHA (ranging from 0.7 g to 3.36 g) could be a promising strategy for improving physical performance in the elderly and consequently for preventing or treating frailty [54].

The innovative aspect of the experimental formula administered in the present study is the use of probiotic LPPS23. To date, there are only a few animal studies that have investigated the effects of LPPS23 on age-related muscle loss. The gut microbiota could affect muscle metabolism, increasing digestion and absorption of certain nutrients and improving energy efficiency [55]. Moreover, a recent study revealed that *L. paracasei* administration increases amino acid absorption from plant protein and that probiotic supplementation can be an important nutritional strategy to improve changes in post-prandial blood amino acids [56]. Furthermore, LPPS23 has been shown to delay some age-related diseases, including decelerating age-related muscle loss. 

The present findings were in agreement with a recent study on aged mice by Chen et al. [35], who first provided evidence that administration of the probiotic LPPS23 for 12 weeks attenuated the progression of sarcopenia by reducing the inflammatory process at the level of the muscle tissue. An improved inflammatory state could therefore be responsible for balancing protein synthesis and degradation, while inflammation is known to be related to the activation of catabolism and the suppression of muscle protein synthesis [57]. 

Another interesting finding of our study is related to the fact that there was an increase in total FM in the intervention group; however, visceral fat was significantly reduced. It is recognized that body weight associated with maximal survival increases with increasing age [58], and subcutaneous fat is known for its protective role [59]. Consequently, an increase in total subcutaneous fat has a positive effect since visceral fat, which is associated with increased levels of inflammatory markers, was reduced. Although total adiposity is strongly associated with metabolic and cardiovascular risk, it is well known that different fat compartments contribute differentially to these risks [60]. The inflammatory status, assessed by levels of CRP, showed a statistically significant decrease in the intervention group compared with the placebo group from baseline to the 60-day timepoint. These findings provide evidence for the simultaneous effect of omega-3 and LPPS23 anti-inflammatory properties, which may have contributed to a lower degree of inflammation in patients with sarcopenia and also confirm in vivo data [35], highlighting that a reduction in the inflammatory state is important in the management of sarcopenia [61].

The correlation between intestinal microbiota composition and muscle function has given rise to the recent concept of the “gut-muscle axis”, through which the microbiome influences muscle structures and functions starting from the intestine, through the regulation of inflammation and reactive oxygen species production, and mitochondrial function in muscle [30].

The food for special medical purposes administered in the present study was found to be safe and well tolerated, with no serious adverse events reported. 

The compliance for diet and physical activity compliance, assessed by visual analogic scale, was high for both groups, and there were no significant differences between the two groups, thus demonstrating that there was no influence of the diet on the results obtained in this study.

There are some limitations that should be considered in the interpretation of the results observed. First, all dropout patients, with the exception of one (who died for other non-treatment-related reasons), abandoned the study due to the unpleasant taste and odor of the food for special medical purposes attributed to its omega-3 fatty acid content. Second, it was not possible to determine the amount of each nutrient when assessing the effectiveness of the food for special medical purposes because the individual nutrients were not evaluated with other arms of the study.

## 5. Conclusions

The findings of this study indicate that the administration of a food for special medical purposes based on omega-3 polyunsaturated fatty acids, leucine, and the probiotic LPPS23 appears to be a valid strategy to counteract the progression of sarcopenia and sarcopenia-defining parameters in older adults.

## Figures and Tables

**Figure 1 nutrients-14-04566-f001:**
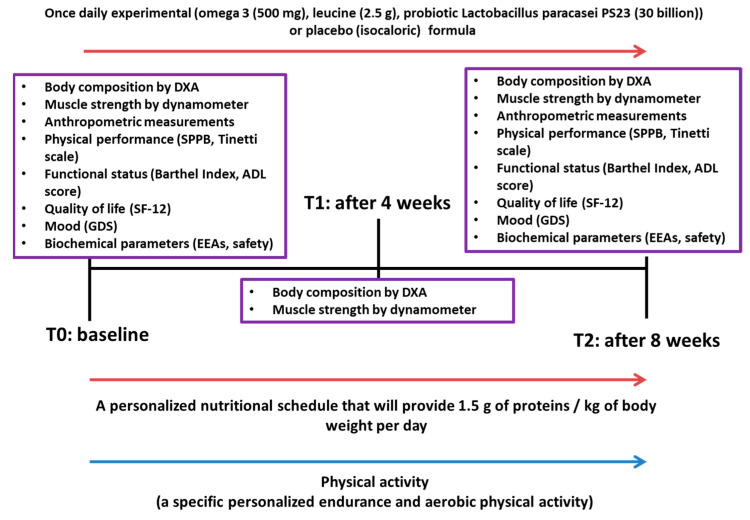
Study Design.

**Figure 2 nutrients-14-04566-f002:**
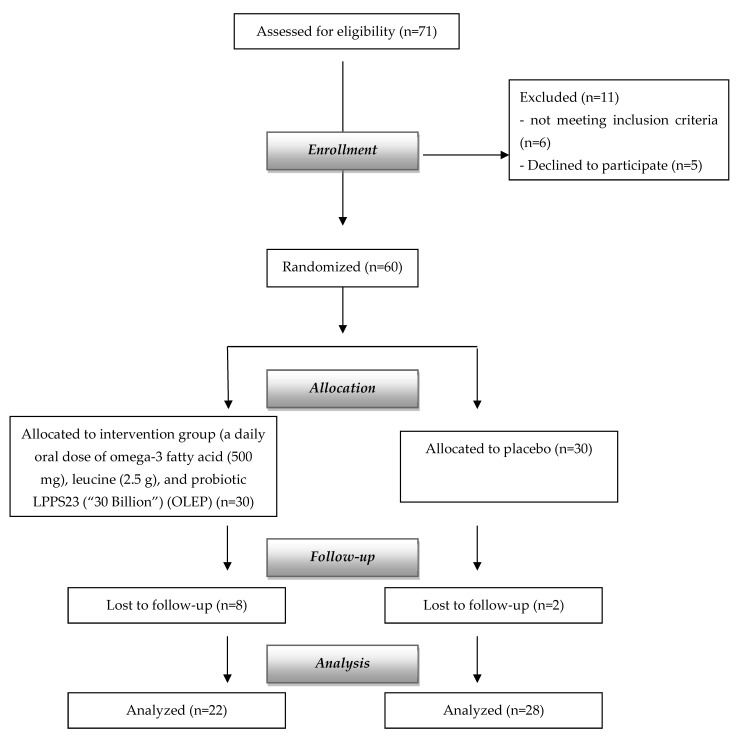
Flowchart of the study.

**Figure 3 nutrients-14-04566-f003:**
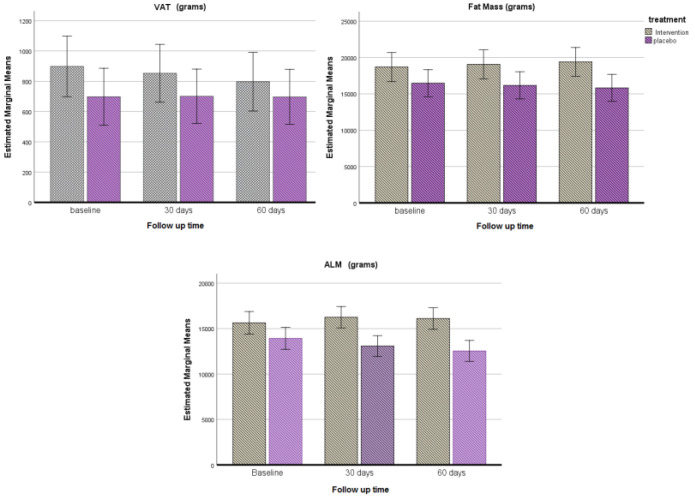
Body composition changes after 30 and 60 days of treatment.

**Figure 4 nutrients-14-04566-f004:**
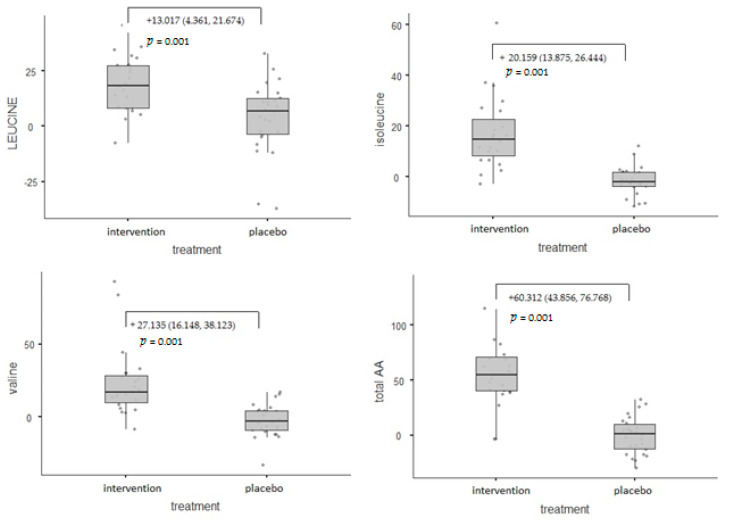
Between-group effects on the amino acids profile.

**Table 1 nutrients-14-04566-t001:** The baseline clinical characteristics of patients between the intervention and placebo groups.

Variable	Intervention Group (*n* = 22)	Placebo Group (*n* = 28)	Total (*n* = 50)	*p*-Value
Age (years)	78.84 ± 5.80	80.50 ± 3.74	79.71 ± 4.84	0.231
Weight (kg)	57.53 ± 11.41	52.32 ± 8.33	54.77 ± 10.14	0.072
BMI (kg/m^2^)	22.63 ± 2.74	21.96 ± 1.34	22.27 ± 2.12	0.271
ALM (kg)	15.646 ± 3.666	13.935 ± 2.020	14.773 ± 3.035	0.052
ALM/h2	6.20 ± 1.23	5.80 ± 0.83	6.00 ± 1.04	0.270
SMI	27.40 ± 4.98	26.61 ± 4.18	26.99 ± 4.60	0.521
FM (g)	18,701.52 ± 4953.04	16,474.92 ± 4516.35	17,520.06 ± 4809.19	0.106
VAT (g)	898.48 ± 648.860	698.04 ± 238.10	792.12 ± 482.40	0.149
WC (cm)	81.22 ± 6.95	79.46 ± 8.14	80.29 ± 7.58	0.424
DBP (mm Hg)	77.30 ± 4.19	78.65 ± 5.08	78.02 ± 4.68	0.319
SBP (mm Hg)	126.96 ± 5.59	125.58 ± 5.89	126.22 ± 5.73	0.406
Tinetti (score)	17.83 ± 4.01	16.88 ± 3.70	17.33 ± 3.84	0.397
SPPB total (score)	4.22 ± 1.45	4.38 ± 0.64	4.31 ± 1.08	0.595
Handgrip (kg)	18.74 ± 4.33	17.64 ± 4.68	18.06 ± 4.52	0.328
Barthel (score)	60.43 ± 18.64	55.58 ± 19.30	57.86 ± 18.96	0.376
ADL (score)	3.22 ± 0.85	3.19 ± 1.50	3.20 ± 1.22	0.944
Physical SF-12 (score)	33.80 ± 6.83	37.29 ± 11.82	35.65 ± 9.86	0.220
Mental SF-12 (score)	40.78 ± 8.73	40.89 ± 12.40	40.83 ± 10.56	0.973
GDS (score)	14.13 ± 1.98	13.88 ± 1.77	14.00 ± 1.86	0.649
Creatinine (mg/dL)	0.86 ± 0.13	0.79 ± 0.21	0.82 ± 0.18	0.181
AST (IU/L)	19.26 ± 8.86	18.04 ± 8.81	18.61 ± 8.77	0.631
ALT (IU/L)	16.17 ± 7.99	16.38 ± 10.87	16.29 ± 9.53	0.939
GGT (IU/L)	15.90 ± 3.15	16.37 ± 2.39	16.15 ± 2.75	0.560
FBG (mg/dL)	86.09 ± 11.16	88.92 ± 9.43	87.59 ± 10.27	0.340
CRP (mg/dL)	1.20 ± 1.45	0.94 ± 0.59	1.06 ± 1.08	0.392
Leucine	105.30 ± 18.33	99.95 ± 24.19	102.46 ± 21.59	0.393
Isoleucine	88.25 ± 24.09	88.63 ± 25.47	88.45 ± 24.58	0.958
Valine	133.30 ± 36.23	143.98 ± 41.39	138.97 ± 39.02	0.344
Total AA	326.85 ± 38.43	332.56 ± 57.64	329.88 ± 49.15	0.689

Abbreviations: AA, amino acid; ADL, activity daily living; ALM, appendicular lean mass; ALT, alanine aminotransferase; AST, aspartate aminotransferase; BMI, body mass index; CRP, C-reactive protein; DBP, diastolic blood pressure; FBG, fasting blood glucose; GDS, geriatric depression scale; GGT, gamma-glutamyl transferase; SPPB, short physical performance battery; SBP, systolic blood pressure; SF-12, 12-item short-form health survey; VAT, visceral adipose tissue; WC, waist circumference.

**Table 2 nutrients-14-04566-t002:** Primary and secondary endpoints.

Variable	Intervention Group (N = 22) Mean Change from Baseline (95% CI)	Placebo Group (N = 28) Mean Change from Baseline (95% CI)	Difference between Groups (95% CI)	*p*-Value between Group
Weight (kg)	1.378 (0.813, −1.943)	−1.442 (−1.972, −0.912) *	2.820 (2.028, 3.611)	<0.001
BMI (kg/m^2^)	0.539 (0.308, 0.769) *	−0.600 (−0.816, −0.384) *	1.139 (0.816, 1.462)	<0.001
ALM (kg)	0.350 (−0.607, 1.307)	−1.268 (−2205.44, −332.26) *	1.618 (−255.75, 2982.00)	0.245
ALM/h2	0.516 (−0.114; 1.146)	−0.505 (−1.103; 0.097)	1.021 (−1.910; −0.132)	<0.05
SMI (%)	0.282 (−1.098; 1.662)	−0.950 (−2.269; 0.368)	1.232 (−0.719; 3.170)	0.208
FM (g)	700.522 (360.688, 1040.356) *	−646.962 (−965.724, −328.200) *	1347.484 (871.386, 1823.582)	<0.001
VAT (g)	−113.171 (−188.061, −38.280) *	9.151 (−61.096, 79.398)	−122.321 (−227.241, −17.402)	<0.001
WC (cm)	0.441 (−0.005, 0.887)	−0.582 (−1.000, −0.164) *	1.023 (0.399, 1.648)	<0.001
DBP (mm Hg)	0.214 (−1.364; 1.764)	−0.574 (−2.055; 0.907)	0.788 (−1.423; 3.000)	0.477
SBP (mm Hg)	−0.332 (−2.004, 1.341)	−0.399 (−1.968, 1.170)	0.067 (−2.276, 2.411)	0.954
Tinetti (score)	1.940 (0.989, 2.891) *	−0.447 (−1.339, 0.446)	2.386 (1.054, 3.719)	<0.001
SPPB total (score)	2.667 (2.108, 3.226) *	0.448 (−0.076, 0.973)	2.219 (1.436, 3.002)	<0.001
Handgrip (kg)	3.332 (2.400, 4.264) *	−0.755 (−1.629, 0.119)	4.087 (2.781, 5.392)	<0.001
Barthel (score)	4.222 (0.480, 7.964) *	0.111 (−3.399, 3.621)	4.111 (−1.132, 9.353)	0.121
ADL (score)	0.506 (0.127, 0.885) *	−0.101 (−0.457, 0.254)	0.607 (0.076, 1.138)	<0.001
Physical SF-12 (score)	2.747 (−0.472, 5.966)	3.519 (0.499, 6.538) *	−0.772 (−5.281, 3.738)	0.732
Mental SF-12 (score)	1.238 (−0.631, 3.107)	3.789 (2.036, 5.543) *	−2.551 (−5.170, 0.067)	0.056
GDS (score)	−0.262 (−0.631, 0.106)	0.155 (−0.190, 0.501)	−0.418 (−0.934, 0.099)	0.110
Creatinine (mg/dL)	−0.025 (−0.078, 0.027)	0.043 (−0.007, 0.092)	−0.068 (−0.141, 0.005)	0.069
AST (IU/L)	0.421 (−2.311, 3.154)	0.435 (−2.128, 2.998)	−0.013 (−3.841, 3.815)	0.994
ALT (IU/L)	3.150 (0.042, 6.258) *	0.059 (−2.856, 2.974)	3.091 (−1.263, 7.445)	0.160
GGT (IU/L)	0.235 (−0.236, 0.705)	0.069 (−0.372, 0.511)	0.165 (−0.494, 0.824)	0.617
FBG (mg/dL)	−0.904 (−5.580, 3.773)	−2.470 (−6.856, 1.917)	1.566 (−4.986, 8.118)	0.633
CRP (mg/dL)	−0.690 (−1.094, −0.286) *	0.266 (−0.113, 0.645)	−0.956 (−1.522, −0.390)	<0.001
Leucine	17.832 (11.653, 24.010) *	4.814 (−0.981, 10.610)	13.017 (4.361, 21.674)	<0.001
Isoleucine	17.855 (13.369, 22.341) *	−2.304 (−6.512, 1.903)	20.159 (13.875, 26.444)	<0.001
Valine	24.020 (16.177, 31.863) *	−3.116 (−10.472, 4.241)	27.135 (16.148, 38.123)	<0.001
Total AA	59.706 (47.960, 71.452) *	−0.606 (−11.623, 10.412)	60.312 (43.856, 76.768)	<0.001

***** *p* value set up at <0.001 Abbreviations: AA, amino acid; ADL, activity daily living; ALM, appendicular lean mass; ALT, alanine aminotransferase; AST, aspartate aminotransferase; BMI, body mass index; CI, confidence interval; CRP, C-reactive protein; DBP, diastolic blood pressure; FBG, fasting blood glucose; GDS, geriatric depression scale; GGT, gamma-glutamyl transferase; SPPB, short physical performance battery; SBP, systolic blood pressure; SF-12, 12-item short-form health survey; VAT, visceral adipose tissue; WC, waist circumference.

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
