# Peer review of "Effectiveness of a Novel Food Composed of Leucine, Omega-3 Fatty Acids and Probiotic Lactobacillus paracasei PS23 for the Treatment of Sarcopenia in Elderly Subjects: A 2-Month Randomized Double-Blind Placebo-Controlled Trial"

_nutrients, 2022, doi:10.3390/nu14214566_

Round 1

Reviewer 1 Report

The manuscript by Rondanelli et al., explores the effects of a 60 day intervention consisting of leucine, omega-3 fatty acids, and a probiotic (Lactobacillus paracasei PS23) in elderly individuals with sarcopenia. The authors report an overall benefit in terms of lean mass, plasma amino acid levels, and other functional outcomes (handgrip strength). and This is an interesting study that has value to the field. However, several points need to be address before the manuscript is suitable for publication. 

Introduction

·      1st paragraph of the introduction needs to be removed as it is only instructions.

Methods

·      Page 3: last word is misspelled. Should be spelled “enrollment”

·      Additional details regarding the source of probiotic and form It was delivered (e.g, freeze dried, etc)

·      What is the composition of the Omega 3 supplement? Specially, DHA vs EPA content.

·      How as the experimental formula delivered? Drink, pill, powder, food, other?

·      How was CRP measured?

Results

·      Table 2: the value for ALM is not reported correctly. I think the correct value should be “0.350”

·      Why is the p value set up to >0.05. The p value should be <0.05. The same goes for pvalue set up to >0.001. The p value for significance at this level should be P <0.0001.

·      The difference in ALM change from baseline compared between intervention and placebo group is not significant as indicated by the CI and P-value. Also, there was a decrease in in ALM in the placebo group compared to baseline. Please revise this.

·      There was a significant decrease in CRP levels in the intervention group compared to baseline, but this is not reflected in Table 2.

·      ALT increased in the intervention group compared to baseline.

·      The graphs in figure 3 are very misleading as the y-axis does not originate at 0. This makes it seem like there is a larger difference that what the data are truly showing. Please update the y-axis. The authors can also convert the ALM and fat mass to kg decrease the values and make it easier to interpret. The graphs are also lacking any type of error bars. These graphs must be revised.

·      Figure 5 does not add any significant value to manuscript.

Discussion

·      ALM was not significantly changed from baseline in the intervention group. This statement is not accurate. It appears that the intervention prevented loss of ALM that was observed in the placebo group.

·      Are the authors able to provide baseline protein, lipid, and carbohydrate intake of the participants?

·      Page 15, line 495: The is an over interpretation of the data. The study design does not allow the authors to conclude that LLPS23 has any anti-inflammatory role in the context of sarcopenia. Omega-3 fatty acids also have anti-inflammatory properties which may have contributed. The authors need to carefully revise this statement.

·      What do the authors conclude regarding the mental and physical SF-12 score increasing in placebo but not in intervention.

Author Response

Introduction

  • 1st paragraph of the introduction needs to be removed as it is only instructions.

ANSWER: The paragraph has been removed, we’re sorry for the inconvenience.

Methods

  • Page 3: last word is misspelled. Should be spelled “enrollment”

ANSWER: The word has been corrected.

  • Additional details regarding the source of probiotic and form It was delivered (e.g, freeze dried, etc)

ANSWER:    Additional details have been added.

  • What is the composition of the Omega 3 supplement? Specially, DHA vs EPA content.

ANSWER: DHA vs EPA content has been specified.

  • How as the experimental formula delivered? Drink, pill, powder, food, other?

ANSWER: the experimental formula has been delivered as a powder to be dissolved in a glass of water. This detail has been added in the method section.

  • How was CRP measured?

ANSWER: details have been added in the method section: “The CRP was determined by Nephelometric High-Sensitivity CRP (Dade Behring, Marburg, Germany)”.

Results

  • Table 2: the value for ALM is not reported correctly. I think the correct value should be “0.350”

ANSWER: We corrected this mistake.

  • Why is the p value set up to >0.05. The p value should be <0.05. The same goes for pvalue set up to >0.001. The p value for significance at this level should be P <0.0001.

ANSWER: we revised the symbol accordingly.

  • The difference in ALM change from baseline compared between intervention and placebo group is not significant as indicated by the CI and P-value. Also, there was a decrease in in ALM in the placebo group compared to baseline. Please revise this.

ANSWER: We revised this statement accordingly.

  • There was a significant decrease in CRP levels in the intervention group compared to baseline, but this is not reflected in Table 2.

ANSWER: We included the * as statistically significant and we corrected this refuse.

  • ALT increased in the intervention group compared to baseline.

ANSWER: we have not focused on this aspect as it has no clinical significance in this case.

  • The graphs in figure 3 are very misleading as the y-axis does not originate at 0. This makes it seem like there is a larger difference that what the data are truly showing. Please update the y-axis. The authors can also convert the ALM and fat mass to kg decrease the values and make it easier to interpret. The graphs are also lacking any type of error bars. These graphs must be revised.

ANSWER: we remade from scratch the histograms with error bars as you suggested. We included the x from 0.

  • Figure 5 does not add any significant value to manuscript.

ANSWER: thank you for the suggestion, we have removed the figure.

Discussion

  • ALM was not significantly changed from baseline in the intervention group. This statement is not accurate. It appears that the intervention prevented loss of ALM that was observed in the placebo group.

ANSWER: The statement has been rearranged as suggested.

  • Are the authors able to provide baseline protein, lipid, and carbohydrate intake of the participants?

ANSWER: As we reported in the method section participants were given an individualized dietary program: “Each participant was given an individualized dietary program. A personalized nutritional schedule was prepared for patients in both groups that provided 1.5 grams of protein per kg of body weight per day. A trained dietitian was responsible for the evaluation of calorie intake with the estimated basal metabolic rate using the Harris-Benedict formula multiplied by the estimated activity factor, and protein in-take.”

Details about lipids and carbohydrate intake have been added.

The compliance to dietary intervention was assessed through the 24 hours-dietary recall once a month for three months.

  • Page 15, line 495: The is an over interpretation of the data. The study design does not allow the authors to conclude that LLPS23 has any anti-inflammatory role in the context of sarcopenia. Omega-3 fatty acids also have anti-inflammatory properties which may have contributed. The authors need to carefully revise this statement.

ANSWER:  Thanks a lot for this suggestion. We rephrased this statement, suggesting that the simultaneous effect of omega 3 and LPPS23 lowered the inflammation in sarcopenic patients.

  • What do the authors conclude regarding the mental and physical SF-12 score increasing in placebo but not in intervention.

ANSWER: SF–12 Intra group pre – post mean difference changes have not been discussed into the paper because they did not affect the between group changes.

The similarities have been corrected and the overlapping sentences have been changed.

Reviewer 2 Report

In their submitted manuscript, Rondanelli et al., report on the effectiveness of a novel supplement to improve markers of sarcopenia in elderly subjects. In this manuscript, the authors found that a supplement containing omega-3 FAs, leucine, and a probiotic, was sufficient to improve clinical measures of sarcopenia in the elderly after a 60-day follow-up. While most my concerns were minor, there was a Major limitation that was either not reported, or not considered by the authors. I would ask that the authors consider all my recommendations below.

Major

1. Although the baseline Weight and ALM differences between groups were reported as not different based on statistical significance, the numerical differences were large and they did approach significance (0.072 and 0.052, respectively). This would suggest that maybe the groups were different, which can happen just based on random chance. If the intervention group had more body weight and ALM, perhaps they had other lifestyle or nutritional backgrounds that would impact their adherence to the nutritional and exercise regimens imposed on all participants of the current study. Since all nutritional and exercise regimens were monitored by professionals, we should be able to gain a better understanding of this. Neither the nutritional adherence, nor the exercise adherence, were reported herein. Can the authors provide detailed breakdowns of subject compliance and its potential impact on the outcome variables reported in the study? For example, the intervention group had increases in plasma AAs across the board, but the placebo group did not. If the nutritional and/or exercise compliance differed between groups then the authors would have to consider this confounding effect in their interpretation of findings and the interpretation of their findings would be vastly different.

2. In addition, ALM alone (especially given variability in body weight) should not be the sole marker of muscle mass loss in sarcopenia, either between groups or over time. It is common to normalize multiple lean mass measures to better reflect other anthropometric variables such as height or weight. The authors should review some of these normalizations and report them together with the raw ALM they have reported:

(Moon et al., doi: 10.11005/jbm.2018.25.1.15) and (Walowski et al., doi: 10.3390/nu12030755)

3. Figure 5 is labeled as correlations for both intervention and placebo groups. However, only 17 points are included (18 from the Handgrip x SPPB correlation), from what I assume is the intervention group as explained in the figure summary. Where are the 28 missing data points from the placebo group? Shouldn’t both groups be represented in this analysis?

Minor

4. Remove the description of the introduction (lines 46-55)

5. Anthropometric measures do not provide an assessment of ‘nutritional status’ (lines 181-182)

6. Please provide a short explanation of the ‘standard procedures’ employed in this study and references to support the validity of the procedures (lines 199-202)

Author Response

Major

  1. Although the baseline Weight and ALM differences between groups were reported as not different based on statistical significance, the numerical differences were large and they did approach significance (0.072 and 0.052, respectively). This would suggest that maybe the groups were different, which can happen just based on random chance. If the intervention group had more body weight and ALM, perhaps they had other lifestyle or nutritional backgrounds that would impact their adherence to the nutritional and exercise regimens imposed on all participants of the current study. Since all nutritional and exercise regimens were monitored by professionals, we should be able to gain a better understanding of this. Neither the nutritional adherence, nor the exercise adherence, were reported herein. Can the authors provide detailed breakdowns of subject compliance and its potential impact on the outcome variables reported in the study? For example, the intervention group had increases in plasma AAs across the board, but the placebo group did not. If the nutritional and/or exercise compliance differed between groups then the authors would have to consider this confounding effect in their interpretation of findings and the interpretation of their findings would be vastly different.

ANSWER: New sentences have been added in the method section. The compliance to dietary intervention was assessed through the 24 hours-dietary recall once a month for three months. For both groups  a specific personalized endurance and aerobic physical activity training has been set.

  1. In addition, ALM alone (especially given variability in body weight) should not be the sole marker of muscle mass loss in sarcopenia, either between groups or over time. It is common to 8normalize multiple lean mass measures to better reflect other anthropometric variables such as height or weight. The authors should review some of these normalizations and report them together with the raw ALM they have reported:

(Moon et al., doi: 10.11005/jbm.2018.25.1.15) and (Walowski et al., doi: 10.3390/nu12030755)

ANSWER: ALM has been standardized by BMI as suggested by the 2 studies (Moon et al, Walowski et al) in order to normalize the data by height and weight. We have entered the analyzed data in table 2, even with standardization the data is not significant. New sentences have been added in the method section and discussion section.

  1. Figure 5 is labeled as correlations for both intervention and placebo groups. However, only 17 points are included (18 from the Handgrip x SPPB correlation), from what I assume is the intervention group as explained in the figure summary. Where are the 28 missing data points from the placebo group? Shouldn’t both groups be represented in this analysis?

ANSWER:  Dear Reviewer, thanks a lot for this question. The spearman correlations have been made only on the intervention group in order to understand the physiological mechanisms of the difference changes over time. As you counted, we included the cohort of supplemented group (total sample 22 subject) with 4 missing data for handgrip. We included this statement below the figures.

Minor

  1. Remove the description of the introduction (lines 46-55)

ANSWER: The paragraph has been removed, we’re sorry for the inconvenience.

  1. Anthropometric measures do not provide an assessment of ‘nutritional status’ (lines 181-182)

ANSWER: The sentence has been modified.

  1. Please provide a short explanation of the ‘standard procedures’ employed in this study and references to support the validity of the procedures (lines 199-202)

ANSWER: A short explanation of the procedure has been provided.

The similarities have been corrected and the overlapping sentences have been changed

Round 2

Reviewer 1 Report

The authors have addressed my concerns. 

Author Response

Thank you for your kind collaboration and suggestions.

Best regards,

The authors

Reviewer 2 Report

The responses to my primary concerns were inadequate. I will outline IN DETAIL, what needs to be included. This is essential so that the reader can make an informed decision and will help ensure that the authors' interpretations are supported by the data.

1. ALM/BMI is NOT an acceptable criterion measure. It was not presented as one in either of the documents that were previously shared. Please remove it from Table 2.

2. Please provide a baseline normalization of (kg ALM/ht2) in Table 1. Include the change in this variable in Table 2.

3. Please provide skeletal muscle mass index (SMI) as a percentage in Table 1. ((kg ALM/kg body mass) x 100). Include the change in this variable in Table 2.

4. A dietary recall does NOT provide an indication of patient compliance. Compliance should be quantifiable, if truly monitored as the authors suggest. For example, the authors should have a number of exercise training sessions completed compared to a number of exercise training sessions scheduled. If someone attended 76 out of 100 possible session, then compliance was 76%. What was the compliance of training for each member of each group? Were the rates of compliance different between groups? Please compare these rates of compliance between groups

Based on the authors explanation concerning nutrition compliance, a daily recall is insufficient. Did they not maintain a food log, or register what days they completed ALL nutrition recommendations?

By not accounting for these potential variables, and given the 'trending' differences in baseline anthropometric measures, the authors are unable to confidently rule out nutrition and diet as the primary contributors to their observed effects.

Author Response

We send you the revised manuscript together with our point-by-point response.

The changes in the text were highlighted in green.

Thanking you in advance for your kind collaboration and suggestions.

Best regards,

The authors

  1. ALM/BMI is NOT an acceptable criterion measure. It was not presented as one in either of the documents that were previously shared. Please remove it from Table 2.

Answer: DONE

  1. Please provide a baseline normalization of (kg ALM/ht2) in Table 1. Include the change in this variable in Table 2.

Answer: DONE.

  1. Please provide skeletal muscle mass index (SMI) as a percentage in Table 1. ((kg ALM/kg body mass) x 100). Include the change in this variable in Table 2.

Answer: DONE.

  1. A dietary recall does NOT provide an indication of patient compliance. Compliance should be quantifiable, if truly monitored as the authors suggest. For example, the authors should have a number of exercise training sessions completed compared to a number of exercise training sessions scheduled. If someone attended 76 out of 100 possible session, then compliance was 76%. What was the compliance of training for each member of each group? Were the rates of compliance different between groups? Please compare these rates of compliance between groups

Based on the authors explanation concerning nutrition compliance, a daily recall is insufficient. Did they not maintain a food log, or register what days they completed ALL nutrition recommendations?

By not accounting for these potential variables, and given the 'trending' differences in baseline anthropometric measures, the authors are unable to confidently rule out nutrition and diet as the primary contributors to their observed effects.

Answer: In the methods and in the results sections, we have added new sentences in order to better clarify how we assessed compliance to diet and physical activity.

A percentage standardization of the visual analogic scale for diet and physical activity compliance was performed. Qualitative variables were described as frequencies (%) with respect to an ideal situation (100% compliance) and the statistical differences were evaluated by Chi-squared tests. The results of this analysis demonstrated that there were no significant differences between the two groups for compliance to diet and physical activity, thus demonstrating that there is no influence of the diet on the results obtained in this study.

Finally, in the discussion a sentence has been added on this topic.